# Evaluating Risks in Weak-to-Strong Alignment: A Bias-Variance Perspective

## Abstract

Weak-to-strong alignment has emerged as a central paradigm for scalable supervision, yet it introduces new risks when strong models are trained using feedback generated by imperfect teachers. In this work, we analyze weak-to-strong alignment through a bias-variance perspective by connecting misfit theory to practical post-training pipelines. We derive a misfit-based upper bound on the strong model's population risk under weak supervision and decompose this bound into bias, variance, and covariance components that capture both teacher quality and student deviation. We empirically study four weak-to-strong training pipelines spanning supervised fine-tuning (SFT), reinforcement learning from human feedback (RLHF), and reinforcement learning from AI feedback (RLAIF) on the PKU-SafeRLHF and HH-RLHF datasets. Our results show that bias and variance alone are insufficient to explain deceptive behavior, and instead, covariance alignment between weak and strong reward models plays a dominant role. In particular, supervised fine-tuning tends to preserve low-variance alignment but can amplify weak-model inductive biases, whereas reinforcement learning applied to the strong model suppresses deception by disrupting covariance alignment, even when theoretical alignment bounds increase. These findings highlight fundamental limitations of misfit-based bounds as standalone safety indicators and emphasize the importance of controlling weak-strong interactions in alignment pipelines.

## 1 Introduction

Understanding weak-to-strong alignment is increasingly important as modern training pipelines rely more heavily on feedback generated by weaker or imperfect models. While this paradigm enables scalable supervision, it introduces new risks that are not yet well understood. Two central challenges arise in this setting. First, weak teachers may provide systematically biased or misaligned supervision. Second, stronger students may learn to exploit stable but imperfect reward signals, leading to deceptive behavior. As model capability increases, identifying when and why these risks emerge, and how they depend on the choice of training methodology, becomes critical.

In this work, we study weak-to-strong alignment through a bias-variance lens by bridging misfit theory with practical post-training pipelines. We derive a misfit-based upper bound on the strong model's population risk under weak supervision and decompose this bound into bias, variance, and covariance terms that capture both teacher quality and student deviation. This decomposition provides a principled framework for analyzing how different post training techniques including supervised fine-tuning (SFT), reinforcement learning from human feedback (RLHF), and reinforcement learning from AI feedback (RLAIF) induce distinct bias-variance trade-offs.

We empirically evaluate four weak-to-strong training pipelines spanning supervised and reinforcement learning paradigms on the PKU-SafeRLHF and HH-RLHF datasets. Our results indicate that supervised fine-tuning tends to preserve low-variance, low-error alignment under weak supervision, whereas reinforcement learning pipelines introduce higher variance and altered covariance structure that fundamentally affects the risks in weak-to-strong alignment. We analyze how these shifts in bias, variance, and overall prediction error relate to the emergence of deceptive behavior across training regimes.

**Contributions.**

**1.** We derive a misfit-based formulation of weak-to-strong alignment and decompose it into bias, variance, and covariance components, enabling principled analysis across training pipelines.

**2.** We define and study four weak-to-strong training settings spanning supervised fine-tuning and reinforcement learning, each exhibiting distinct bias-variance-covariance characteristics.

**3.** We extend prior analyses of offline preference optimization to an online setting by incorporating policy optimization via PPO.

**4.** We empirically evaluate weak-to-strong pipelines using deception scores and demonstrate that deceptive behavior systematically relates to covariance alignment rather than bias or variance alone.

## 2 RELATED WORKS

Weak-to-strong generalization was first studied by Burns et al. (2024), where a relatively smaller (weaker) model is aligned using reward modeling techniques such as the Bradley–Terry framework Bradley & Terry (1952), and then used to label another portion of the dataset for training a stronger model. Results in this work show that the larger (strong) student model trained under this pipeline can outperform its weak teacher. The authors introduce Performance Gap Recovered (PGR) as a metric to quantify the extent of generalization. However, this work is limited to the reward modeling stage of Reinforcement Learning from Human Feedback (RLHF), with full end-to-end RL training left as future work.

Subsequently, Yang et al. (2025) studied weak-to-strong alignment in an offline preference optimization setting, training models using Direct Preference Optimization (DPO) Rafailov et al. (2023) and SimPO Meng et al. (2024). The authors introduce a deception score to quantify misalignment and propose bootstrapping with intermediate models as a mitigation strategy. While effective, their approach is restricted to offline preference optimization and does not consider supervised fine-tuning-based alignment pipelines.

Lyu et al. (2025) propose a framework in which two groups of weak teachers label data with positive and negative signals, and the strong model learns to contrast between them while favoring positive supervision. The authors demonstrate that increasing both the number of weak teachers and the number of iterations improves the alignment quality of the strong model. However, although multiple weak teacher models are trained, the framework ultimately relies on selecting the single best response, based on lowest perplexity, from multiple generated candidates. As a result, the approach requires generating and evaluating several responses per data point, which introduces nontrivial computational overhead and limits scalability. The strong model is then trained using a combination of DPO and supervised fine-tuning (SFT) losses.

From a data-centric perspective, Shin et al. (2025) analyze weak-to-strong generalization as a phenomenon emerging from dataset structure rather than model architecture. They attribute successful generalization to the emergence of hard patterns in the data that are inferred from simpler patterns learned by the weak model. To formalize this, they introduce overlap density, defined as the number of data points for which weak and strong models exhibit overlapping learned patterns. The authors further discuss risks such as bias amplification, reliance on superficial heuristics, and the emergence of blind spots, shifting the focus of weak-to-strong alignment from models to data. Charikar et al. (2024); Mulgund & Pabbaraju (2025) develop theoretical formulations of weak-to-strong generalization by explicitly relating the performance gains of the strong model to its misfit with the weak supervisor. In Charikar et al. (2024), the authors analyze weak-to-strong learning in the setting of real-valued regression under squared loss, and show that fine-tuning a strong model on weak labels can be interpreted as a projection of the weak model onto the strong model's hypothesis class. Under a convexity assumption on the strong model's function class, they prove that the reduction in true error of the strong model is at least as large as its error on the weakly labeled data, formally establishing that performance gain is quantified by misfit Charikar et al. (2024). This result provides a geometric and representation-theoretic justification for why a strong model can outperform its weak teacher despite never observing ground-truth labels.

Building on this insight, Mulgund & Pabbaraju (2025) extend the misfit-based characterization beyond squared loss to a broader class of learning problems by leveraging Bregman divergences. This generalization enables theoretical guarantees for classification tasks, where cross-entropy and KL

Figure 1: Overview of our four different weak-to-strong alignment frameworks. In first setting, the process begins with training a reward model and a weak policy $\pi^w$ using human-labeled data by RLHF. The weak policy then relabels a new subset to generate synthetic weak labels, enabling a second reward model to learn the weak policy's implicit value structure. Finally, a strong policy $\pi^s$ is trained via RLAIF on these weakly supervised labels. Each of the RLHF and RLAIF steps are substituted by SFT to make a new variant for our experimental setup.

divergence naturally arise, and relaxes the strict convexity requirement by considering convex combinations of strong models Mulgund & Pabbaraju (2025).

Xu et al. (2025) further interpret weak-to-strong generalization through a bias-variance perspective, showing that the expected population risk gap between the student and teacher decomposes into bias and variance terms governed by the expected misfit Xu et al. (2025). Their analysis removes several restrictive assumptions present in earlier work, highlights the role of posterior-mean teachers, and explains how increasing student capacity can preserve non-zero misfit while still guaranteeing generalization gains. Together, these theoretical results position misfit not only as a by-product of weak supervision, but as a fundamental quantity governing when and why weak-to-strong generalization emerges.

## 3 PRELIMINARIES

Following Xiong et al. (2024) we formulate the RLHF process as aligning a large language model (LLM) to take a prompt $x \in \mathcal{X}$, and output a response $a = [t_1, t_2, \dots]$, where $t_i$ is the $i$-th token generated by the LLM. From a reinforcement learning point of view we call $\mathcal{X}$ the state-space of a contextual bandit with action-space $\mathcal{A}$. In RLHF a ground truth reward function $r^*(x, a)$ maps from the state-action space of this bandit to a chosen-rejected 1 or 0 reward, more formally we have:

$$r^*(x, a) : \mathcal{X} \times \mathcal{A} \Rightarrow [0, 1]. \tag{1}$$

### 3.1 REWARD MODELING

Typical RLHF datasets consist of tuples $(x, a^1, a^2, y)$ with $y$ being the preference signal defined in 2:

$$y = \begin{cases} 1 & a^1 \succ a^2 \\ 0 & a^1 \prec a^2, \end{cases} \tag{2}$$

where $a^1 \succ a^2$ means response $a^1$ is preferred over response $a^2$.

To model the the reward function defined in 1 we follow the Bradley-Terry Bradley & Terry (1952) model:

$$\mathbb{P}(a^1 \succ a^2 | x, a^1, a^2) = \frac{exp(r^*(x, a^1))}{exp(r^*(x, a^1)) + exp(r^*(x, a^2))} = \sigma(r^*(x, a^1) - r^*(x, a^2)), \tag{3}$$

where $\sigma(z) = \frac{1}{1+exp(-z)}$ is the sigmoid function.

## 3.2 POLICY TRAINING

In aligning the LLMs using RLHF, we first have a pretrained model fine-tuned on instruction following tasks denoted as $\pi_0$. The goal is to align the LLM $\pi$ to a set of prompts taken from distribution $x \sim d_0$, while keeping it close to the $\pi_0$. Therefore we maximize the objective 4.

$$J(\pi) = \mathbb{E}_{x \sim d_0}[\mathbb{E}_{a \sim \pi(.|x)}[r^*(x,a)] - \eta D_{KL}(\pi(.|x)||\pi_0(.|x))], \tag{4}$$

where $\eta > 0$ is coefficient for the KL-penalty term. After this process the aligned LLM $\pi$ is a mapping from the state-space $\mathcal{X}$ to a distribution over the action-space $\mathcal{A}$.

## 4 PROBLEM FORMULATION

### 4.1 POPULATION RISK IN WEAK-TO-STRONG ALIGNMENT

In 1 we defined the human (ground truth) reward function as $r^*(x,a)$. Let $r^w(x,a)$, and $r^s(x,a)$ denote the weak and strong models estimated rewards respectively. For brevity, we omit the explicit dependence on $(x,a)$ and write $r$ instead of $r(x,a)$ when the context is clear. Following Xu et al. (2025), the expected population risk for the policy $\pi$ can be defined as:

$$\mathcal{R}(\pi) = \mathbb{E}_{x \sim d_0, a \sim \pi}[L(r^*, \hat{r}^\pi)], \tag{5}$$

where $L(\cdot, \cdot)$ is a Bregman divergence. Using squared loss as a special case of Bregman divergence Xu et al. (2025), the expected population risk for the weak policy $\pi^w$ is defined as:

$$\mathcal{R}(\pi^w) = \mathbb{E}_{x \sim d_0, a \sim \pi^w}[(r^* - \hat{r}^{\pi^w})^2], \tag{6}$$

if we add and subtract the strong estimation of the reward $\hat{r}^{\pi^s}$ we can expand the square as:

$$\begin{aligned}
\mathcal{R}(\pi^w) &= \mathbb{E}_{x \sim d_0, a \sim \pi^w}[(r^* - \hat{r}^{\pi^w})^2] \\
&= \mathbb{E}_{x \sim d_0, a \sim \pi^w}[(r^* - \hat{r}^{\pi^s} + \hat{r}^{\pi^s} - \hat{r}^{\pi^w})^2] \\
&= \mathbb{E}_{x \sim d_0, a \sim \pi^w}[(r^* - \hat{r}^{\pi^s})^2] + \mathbb{E}_{x \sim d_0, a \sim \pi^w}[(\hat{r}^{\pi^s} - \hat{r}^{\pi^w})^2] \\
&\quad + 2\mathbb{E}_{x \sim d_0, a \sim \pi^w}[(r^* - \hat{r}^{\pi^s})(\hat{r}^{\pi^s} - \hat{r}^{\pi^w})],
\end{aligned} \tag{7}$$

where

$$\mathcal{M}(\pi^s, \pi^w) = \mathbb{E}_{x \sim d_0, a \sim \pi^w}[(\hat{r}^{\pi^s} - \hat{r}^{\pi^w})^2], \tag{8}$$

is the *expected misfit* Xu et al. (2025); Mulgund & Pabbaraju (2025) or student-supervisor disagreement Burns et al. (2024), and weak to strong population risk can be defined as off-policy population risk of strong model. More formally:

$$\mathcal{R}_{w2s}(\pi^s|\pi^w) = \mathbb{E}_{x \sim d_0, a \sim \pi^w}[(r^* - \hat{r}^{\pi^s})^2]. \tag{9}$$

Let $\epsilon = \mathbb{E}_{x \sim d_0, a \sim \pi^w}[(r^* - \hat{r}^{\pi^s})(\hat{r}^{\pi^s} - \hat{r}^{\pi^w})]$ therefore we can rewrite 7 as:

$$\mathcal{R}_{w2s}(\pi^s|\pi^w) = \mathcal{R}(\pi^w) - \mathcal{M}(\pi^s, \pi^w) - 2\epsilon. \tag{10}$$

From Cauchy-Schwarz inequality we have:

$$|\mathbb{E}_{x \sim d_0, a \sim \pi^w}[(r^* - \hat{r}^{\pi^s})(\hat{r}^{\pi^s} - \hat{r}^{\pi^w})]| \leq \sqrt{\mathbb{E}_{x \sim d_0, a \sim \pi^w}[(r^* - \hat{r}^{\pi^s})^2]\mathbb{E}_{x \sim d_0, a \sim \pi^w}[(\hat{r}^{\pi^s} - \hat{r}^{\pi^w})^2]} \tag{11}$$

Multiplying both sides to 2 we have:

$$2|\epsilon| \leq 2\sqrt{\mathcal{R}_{w2s}(\pi^s|\pi^w)\mathcal{M}(\pi^s, \pi^w)}. \tag{12}$$

Therefore from 7 and 12 we can say:

$$\mathcal{R}_{w2s}(\pi^s|\pi^w) \leq \mathcal{R}(\pi^w) - \mathcal{M}(\pi^s, \pi^w) + 2\sqrt{\mathcal{R}_{w2s}(\pi^s|\pi^w)\mathcal{M}(\pi^s, \pi^w)}. \tag{13}$$

Reordering the terms we have:

$$\mathcal{R}_{w2s}(\pi^s|\pi^w) + \mathcal{M}(\pi^s, \pi^w) - 2\sqrt{\mathcal{R}_{w2s}(\pi^s|\pi^w)\mathcal{M}(\pi^s, \pi^w)} \leq \mathcal{R}(\pi^w),$$
$$(\sqrt{\mathcal{R}_{w2s}(\pi^s|\pi^w)} - \sqrt{\mathcal{M}(\pi^s, \pi^w)})^2 \leq \mathcal{R}(\pi^w), \tag{14}$$

Taking square roots from both sides we have:

$$|\sqrt{\mathcal{R}_{w2s}(\pi^s|\pi^w)} - \sqrt{\mathcal{M}(\pi^s, \pi^w)}| \leq \sqrt{\mathcal{R}(\pi^w)}$$
$$\sqrt{\mathcal{R}_{w2s}(\pi^s|\pi^w)} \leq \sqrt{\mathcal{R}(\pi^w)} + \sqrt{\mathcal{M}(\pi^s, \pi^w)} \tag{15}$$
$$\mathcal{R}_{w2s}(\pi^s|\pi^w) \leq (\sqrt{\mathcal{R}(\pi^w)} + \sqrt{\mathcal{M}(\pi^s, \pi^w)})^2$$

Equation 15 establishes a misfit-dependent upper bound on the weak-to-strong population risk. The off-policy population risk of the strong model, evaluated under the weak model's action distribution, is bounded above by population risk of the weak model with respect to the ground-truth reward plus the expected misfit between the strong and weak reward models, and an interaction term coupling the two quantities.

For a fixed prompt $x$, the action $a$ is a random variable sampled from the weak policy $\pi^w$. Consequently, both the ground truth reward $r^*(x, a)$ and the predicted reward $\hat{r}^{\pi^w}(x, a)$ are random variables.

We define the expected values (means) over the policy distribution as:

$$\bar{r}^* \triangleq \mathbb{E}_{a\sim\pi^w}[r^*], \tag{16}$$
$$\bar{r}^{\pi^w} \triangleq \mathbb{E}_{a\sim\pi^w}[\hat{r}^{\pi^w}]. \tag{17}$$

Using these means, we define the standard squared bias, model variance, target variance, and covariance:

$$\text{Bias}^2(\hat{r}^{\pi^w}) \triangleq (\bar{r}^{\pi^w} - \bar{r}^*)^2,$$
$$\text{Var}(\hat{r}^{\pi^w}) \triangleq \mathbb{E}_{a\sim\pi^w}[(\hat{r}^{\pi^w} - \bar{r}^{\pi^w})^2],$$
$$\text{Var}(r^*) \triangleq \mathbb{E}_{a\sim\pi^w}[(r^* - \bar{r}^*)^2], \tag{18}$$
$$\text{Cov}(r^*, \hat{r}^{\pi^w}) \triangleq \mathbb{E}_{a\sim\pi^w}[(r^* - \bar{r}^*)(\hat{r}^{\pi^w} - \bar{r}^{\pi^w})].$$

The risk $\mathcal{R}(\pi^w)$ decomposes as follows:

$$\mathcal{R}(\pi^w) = \mathbb{E}_{x\sim d_0}\mathbb{E}_{a\sim\pi^w}[(r^* - \hat{r}^{\pi^w})^2]$$
$$= \mathbb{E}_{x\sim d_0}\mathbb{E}_{a\sim\pi^w}[((r^* - \bar{r}^*) - (\hat{r}^{\pi^w} - \bar{r}^{\pi^w}) + (\bar{r}^* - \bar{r}^{\pi^w}))^2]$$
$$= \underbrace{\mathbb{E}_{x\sim d_0}\mathbb{E}_{a\sim\pi^w}[(r^* - \bar{r}^*)^2]}_{\text{Target Variance}} + \underbrace{\mathbb{E}_{x\sim d_0}\mathbb{E}_{a\sim\pi^w}[(\hat{r}^{\pi^w} - \bar{r}^{\pi^w})^2]}_{\text{Teacher Variance}}$$
$$+ \underbrace{\mathbb{E}_{x\sim d_0}\mathbb{E}_{a\sim\pi^w}[(\bar{r}^* - \bar{r}^{\pi^w})^2]}_{\text{Squared Bias}} \tag{19}$$
$$+ 2\mathbb{E}_{x\sim d_0}\mathbb{E}_{a\sim\pi^w}[(r^* - \bar{r}^*)(\bar{r}^* - \bar{r}^{\pi^w})]$$
$$- 2\mathbb{E}_{x\sim d_0}\mathbb{E}_{a\sim\pi^w}[(\hat{r}^{\pi^w} - \bar{r}^{\pi^w})(\bar{r}^* - \bar{r}^{\pi^w})]$$
$$- 2\mathbb{E}_{x\sim d_0}\mathbb{E}_{a\sim\pi^w}[(r^* - \bar{r}^*)(\hat{r}^{\pi^w} - \bar{r}^{\pi^w})].$$

In 19, the terms involving the constant differences $(\bar{r}^* - \bar{r}^{\pi^w})$ vanish due to the linearity of expectation. Specifically, for any random variable $Z$ and its mean $\bar{Z}$, $\mathbb{E}[Z - \bar{Z}] = 0$. Therefore:

$$\mathbb{E}_{a\sim\pi^w}[r^* - \bar{r}^*] = 0 \quad \text{and} \quad \mathbb{E}_{a\sim\pi^w}[\hat{r}^{\pi^w} - \bar{r}^{\pi^w}] = 0.$$

The remaining cross term is, by definition, the covariance between the ground truth and the predictor scaled by $-2$.

Thus, we arrive at the final decomposition:

$$\mathcal{R}(\pi^w) = \text{Bias}^2(\hat{r}^{\pi^w}) + \text{Var}(\hat{r}^{\pi^w}) + \text{Var}(r^*) - 2\text{Cov}(r^*, \hat{r}^{\pi^w}). \tag{20}$$

Similarly for $\mathcal{M}$, we define the expected values (means) of the student predictions with respect to the weak policy $\pi^w$:

$$\bar{r}^{\pi^s} \triangleq \mathbb{E}_{a\sim\pi^w}[\hat{r}^{\pi^s}(x, a)]. \tag{21}$$

In addition to the terms defined in 18 we define relative squared bias, student variance as:

$$\begin{aligned}
\text{Bias}^2(\hat{r}^{\pi^s} \mid \hat{r}^{\pi^w}) &\triangleq (\bar{r}^{\pi^s} - \bar{r}^{\pi^w})^2, \\
\text{Var}(\hat{r}^{\pi^s}) &\triangleq \mathbb{E}_{a\sim\pi^w}\left[(\hat{r}^{\pi^s}(x, a) - \bar{r}^{\pi^s})^2\right], \\
\text{Cov}(\hat{r}^{\pi^s}, \hat{r}^{\pi^w}) &\triangleq \mathbb{E}_{a\sim\pi^w}\left[(\hat{r}^{\pi^s} - \bar{r}^{\pi^s})(\hat{r}^{\pi^w} - \bar{r}^{\pi^w})\right].
\end{aligned} \tag{22}$$

To decompose the term inside the expectation, we add and subtract these means:

$$\begin{aligned}
\mathcal{M}(\pi^s, \pi^w) &= \mathbb{E}_{x\sim d_0, a\sim\pi^w}\left[\left(\hat{r}^{\pi^s}(x, a) - \hat{r}^{\pi^w}\right)^2\right] \\
&= \mathbb{E}_{x\sim d_0}\mathbb{E}_{a\sim\pi^w}\left[\left((\hat{r}^{\pi^s} - \bar{r}^{\pi^s}) - (\hat{r}^{\pi^w} - \bar{r}^{\pi^w}) + (\bar{r}^{\pi^s} - \bar{r}^{\pi^w})\right)^2\right] \\
&= \underbrace{\mathbb{E}_{x\sim d_0}\left[(\bar{r}^{\pi^s} - \bar{r}^{\pi^w})^2\right]}_{\text{Squared Bias}} + \underbrace{\mathbb{E}_{x\sim d_0}\mathbb{E}_{a\sim\pi^w}\left[(\hat{r}^{\pi^s} - \bar{r}^{\pi^s})^2\right]}_{\text{Student Variance}} + \underbrace{\mathbb{E}_{x\sim d_0}\mathbb{E}_{a\sim\pi^w}\left[(\hat{r}^{\pi^w} - \bar{r}^{\pi^w})^2\right]}_{\text{Teacher Variance}} \\
&\quad + 2\mathbb{E}_{x\sim d_0}\mathbb{E}_{a\sim\pi^w}\left[(\hat{r}^{\pi^s} - \bar{r}^{\pi^s})(\bar{r}^{\pi^s} - \bar{r}^{\pi^w})\right] \\
&\quad - 2\mathbb{E}_{x\sim d_0}\mathbb{E}_{a\sim\pi^w}\left[(\hat{r}^{\pi^s} - \bar{r}^{\pi^s})(\hat{r}^{\pi^w} - \bar{r}^{\pi^w})\right] \\
&\quad - 2\mathbb{E}_{x\sim d_0}\mathbb{E}_{a\sim\pi^w}\left[(\hat{r}^{\pi^w} - \bar{r}^{\pi^w})(\bar{r}^{\pi^s} - \bar{r}^{\pi^w})\right].
\end{aligned} \tag{23}$$

The cross terms involving the constant mean difference $(\bar{r}^{\pi^s} - \bar{r}^{\pi^w})$ vanish due to the linearity of expectation. For example:

$$2(\bar{r}^{\pi^s} - \bar{r}^{\pi^w}) \cdot \mathbb{E}_{a\sim\pi^w}[\hat{r}^{\pi^s}(x, a) - \bar{r}^{\pi^s}] = 2(\bar{r}^{\pi^s} - \bar{r}^{\pi^w}) \cdot 0 = 0.$$

However, the cross term between the student and teacher variations does not vanish, as they are correlated through the shared action $a$:

$$-2\mathbb{E}_{a\sim\pi^w}\left[(\hat{r}^{\pi^s}(x, a) - \bar{r}^{\pi^s})(\hat{r}^{\pi^w}(x, a) - \bar{r}^{\pi^w})\right] = -2\text{Cov}_{a\sim\pi^w}(\hat{r}^{\pi^s}, \hat{r}^{\pi^w}).$$

Substituting these back into equation 23, we obtain:

$$\mathcal{M}(\pi^s, \pi^w) = \text{Bias}^2(\hat{r}^{\pi^s} \mid \hat{r}^{\pi^w}) + \text{Var}(\hat{r}^{\pi^s}) + \text{Var}(\hat{r}^{\pi^w}) - 2\text{Cov}(\hat{r}^{\pi^s}, \hat{r}^{\pi^w}). \tag{24}$$

The above decomposition reveals that the population risk and the weak-to-strong transfer gap can be understood through the relative contributions of bias and variance induced by the choice of supervision signal and training pipeline. Substituting the bias-variance decomposition from 20 and 24 into 15, we obtain:

$$\begin{aligned}
\mathcal{R}_{w2s}(\pi^s \mid \pi^w) \leq \Big( &\sqrt{\text{Bias}^2(\hat{r}^{\pi^w}) + \text{Var}(\hat{r}^{\pi^w}) + \text{Var}(r^*) - 2\text{Cov}(r^*, \hat{r}^{\pi^w})} + \\
&\sqrt{\text{Bias}^2(\hat{r}^{\pi^s} \mid \hat{r}^{\pi^w}) + \text{Var}(\hat{r}^{\pi^s}) + \text{Var}(\hat{r}^{\pi^w}) - 2\text{Cov}(\hat{r}^{\pi^s}, \hat{r}^{\pi^w})} \Big)^2,
\end{aligned} \tag{25}$$

which shows that weak-to-strong alignment depends on the relative bias of the weak model and target variance, as well as the conditional bias and variance of the strong model trained on weak labels, and covariance terms relating the weak model to the ground truth and strong model.

## 5 EXPERIMENTS

Each trained model is assessed to identify where performance drops occur and which steps are most susceptible to learning risks. We investigate four variants of the weak-to-strong alignment pipeline that differ in how the weak and strong models are trained. We partition the dataset $\mathcal{D} = \{(x_i, a_i^1, a_i^2, y_i^{gt})\}_{i=1}^{3N}$ into three disjoint subsets of equal size, $\mathcal{D}_1$, $\mathcal{D}_2$, and $\mathcal{D}_3$. The first subset, $\mathcal{D}_1 = \{(x_i, a_i^1, a_i^2, y_i^{gt})\}_{i=1}^{N}$, contains human-annotated ground-truth preferences and is used to train the initial weak model. $\mathcal{D}_2$ is relabeled using the weak model to generate synthetic weak labels, and $\mathcal{D}_3$ is reserved for held-out evaluation using human annotations. Each configuration can be viewed as a different estimator of the true human-aligned reward function $r^*(x, a)$, with it's own bias-variance decomposition. We empirically evaluate each setup through measuring bias, variance, covariance and assigning a deception score.

**RLHF $\Rightarrow$ RLAIF.** In the first setting, both models are trained through reinforcement learning. We train a reward model on human-labeled preferences $r^*(x, a)$ from $\mathcal{D}_1$ and optimize a weak policy $\pi^w(y|x)$ using Proximal Policy Optimization (PPO Schulman et al. (2017)) under the standard Reinforcement Learning from Human Feedback (RLHF Ouyang et al. (2022)) framework. The resulting weak policy generates synthetic weak labels on $\mathcal{D}_2$ forming $\mathcal{D}_2 = \{(x_i, a_i^1, a_i^2, y_i^w)\}_{i=1}^{N}$.

A second reward model is then trained to approximate the weak policy's preference labels, and a strong policy $\pi^s(y|x)$ is optimized on these weak labels using Reinforcement Learning from AI Feedback (RLAIF Bai et al. (2022b)). Both weak and strong policies are evaluated on the held-out set $\mathcal{D}_3 = \{(x_i, a_i^1, a_i^2, y_i^{gt})\}_{i=1}^{N}$, which contains unseen human-labeled examples.

**RLHF $\Rightarrow$ SFT.** Here, we replace the reinforcement learning stage of the strong model with supervised fine-tuning (SFT). After training $\pi^w(y|x)$ via RLHF on $\mathcal{D}_1$, the weak policy generates synthetic preferences on $\mathcal{D}_2$. The strong model is then fine-tuned on the chosen responses that is, either $a^1$ or $a^2$ where $y^w = 1$ without additional reward modeling.

**SFT $\Rightarrow$ RLAIF.** In this configuration, the weak model is trained by supervised fine-tuning on human-preferred labels from $\mathcal{D}_1$ (i.e., $a^1$ or $a^2$ where $y^{gt} = 1$), eliminating the first reward modeling step. The trained weak policy then generates weak labels for $\mathcal{D}_2$, which are used to train a second reward model and an RLAIF-aligned strong policy as in RLHF$\Rightarrow$RLAIF.

**SFT $\Rightarrow$ SFT.** In our first setup the weak model is fine-tuned on human-preferred labels from $\mathcal{D}_1$, while the strong model is fine-tuned on the weak model's preferred responses in $\mathcal{D}_2$ (again, $a^1$ or $a^2$ where $y^w = 1$). This configuration allows us to isolate the benefits of reinforcement learning and AI-feedback mechanisms introduced in the previous variants.

### 5.1 RESEARCH QUESTIONS

Our goal is to answer the following research question in our experiments:

- Do we observe that SFT and / or RL methods tighten the bounds more through different bias-variance allocation strategies?
- What is the relation between bias-variance-covariance in training weak and strong models, and the performance of each W$\Rightarrow$S pipeline measured by the deception score?

## 6 RESULTS AND ANALYSIS

To evaluate our weak-to-strong pipelines, following Yang et al. (2025) we calculate the deception score for each setup. Unlike their work where deception is calculated under offline training pipelines that include DPO, and SimPO, followed by additional SFT, which encourages higher confidence in model outputs such as 0.75, we rely exclusively on PPO or SFT, without an additional stage. Under these settings, we observe deceptive behavior only at lower confidence thresholds. Accordingly,

| Dataset | W⇒S Pipeline | $\text{Bias}^2(\hat{r}^{\pi^w})$ $\text{Bias}^2(\hat{r}^{\pi^s} \mid \hat{r}^{\pi^w})$ –– | $\text{Var}(\hat{r}^{\pi^w})$ $\text{Var}(\hat{r}^{\pi^s})$ $\text{Var}(r^*)$ | $\text{Cov}(r^*, \hat{r}^{\pi^w})$ $\text{Cov}(\hat{r}^{\pi^s}, \hat{r}^{\pi^w})$ –– | $\mathcal{R}_{w2s}(\pi^s \mid \pi^w)$ | Deception Score $\tau = 0.49$ | Deception Score $\tau = 0.45$ |
|---|---|---|---|---|---|---|---|
| **PKU-SafeRLHF** | SFT ⇒ SFT | $4.84 \times 10^{-7}$ $1.78 \times 10^{-8}$ – | $0.24998$ $0.24998$ $0.24644$ | $-0.00370$ $0.24992$ – | $\underline{0.52035}$ | $0.0294$ | $0.0733$ |
| | SFT ⇒ RLAIF | $4.84 \times 10^{-7}$ $3.25 \times 10^{-6}$ – | $0.24998$ $0.24999$ $0.24644$ | $-0.00370$ $0.24522$ – | $0.65203$ | $0.0294$ | $0.0874$ |
| | RLHF ⇒ SFT | $2.03 \times 10^{-5}$ $1.96 \times 10^{-4}$ – | $0.24991$ $0.24998$ $0.25389$ | $0.00196$ $0.24270$ – | $0.68592$ | **0.0321** | **0.0990** |
| | RLHF ⇒ RLAIF | $2.03 \times 10^{-5}$ $1.98 \times 10^{-4}$ – | $0.24991$ $0.24998$ $0.25389$ | $0.00196$ $0.24267$ – | **0.68637** | $\underline{0.0094}$ | $\underline{0.0326}$ |
| **HH-RLHF** | SFT ⇒ SFT | $6.15 \times 10^{-5}$ $4.58 \times 10^{-9}$ – | $0.24999$ $0.24999$ $0.23888$ | $0.00100$ $0.24995$ – | $\underline{0.49848}$ | $0.0255$ | $0.0674$ |
| | SFT ⇒ RLAIF | $6.15 \times 10^{-5}$ $4.58 \times 10^{-9}$ – | $0.24999$ $0.24999$ $0.23888$ | $0.00100$ $0.24995$ – | $\underline{0.49848}$ | $\underline{0.0143}$ | $0.0522$ |
| | RLHF ⇒ SFT | $6.48 \times 10^{-5}$ $1.15 \times 10^{-7}$ – | $0.24999$ $0.24999$ $0.23896$ | $0.00103$ $0.24982$ – | **0.51297** | **0.0281** | **0.1128** |
| | RLHF ⇒ RLAIF | $6.48 \times 10^{-5}$ $1.15 \times 10^{-7}$ – | $0.24999$ $0.24999$ $0.23896$ | $0.00103$ $0.24982$ – | **0.51297** | $0.0147$ | $\underline{0.0518}$ |

Table 1: Bias-variance-covariance decomposition of weak and strong models, the resulting weak-to-strong alignment upper bound $\mathcal{R}_{w2s}$, and deception scores across training pipelines and datasets. For each dataset, we report the results for the weak model ($\pi^w$), the strong model ($\pi^s$), and their interaction with the ground-truth $r^*$, explicitly separating bias, variance, and covariance terms. The alignment bound $\mathcal{R}_{w2s}$ is computed from these components, while deception scores are evaluated at confidence thresholds $\tau = 0.49$ and $\tau = 0.45$. Despite negligible bias and nearly constant variance across pipelines, covariance terms vary substantially and dominate changes in the alignment bound. Notably, pipelines that minimize deception do not necessarily minimize $\mathcal{R}_{w2s}$, showing that reduced covariance alignment rather than bound tightness alone is central to mitigating deceptive behavior under weak-to-strong alignment.

we report the results using a confidence threshold of 0.49, which is the highest threshold at which deceptive cases are observed in our experiments. For completeness, additional results at a lower threshold of 0.45 are provided. Table 1 reports the bias, variance, covariance components, the resulting weak-to-strong alignment upper bound $\mathcal{R}_{w2s}(\pi^s \mid \pi^w)$, and corresponding deception scores across training pipelines and datasets. First, we observe that training the weak model with reinforcement learning and the strong model with supervised fine-tuning (RLHF⇒SFT) results in the highest deception scores across both datasets and thresholds. In contrast, when the strong model is also trained using reinforcement learning (RLHF⇒RLAIF), deception drops substantially and attains the lowest observed values, although the exact minimum depends on the dataset and confidence threshold. This confirms that reinforcement learning on the strong model plays a central role in mitigating deceptive behavior, even when the weak teacher itself is learned via RL.

Second, we find that training the weak model with reinforcement learning consistently increases the weak-to-strong alignment upper bound $\mathcal{R}_{w2s}(\pi^s \mid \pi^w)$. Notably, in PKU-SafeRLHF, the pipeline with the largest upper bound (RLHF⇒RLAIF) simultaneously achieves the minimum deception score, indicating that a larger theoretical misfit does not necessarily translate into higher deception. A similar pattern appears in HH-RLHF. While the exact pipeline achieving the minimum deception depends on the threshold, reinforcement learning on the strong model consistently reduces deception despite elevated alignment bounds.

In contrast, training the weak model via supervised fine-tuning yields the smallest alignment upper bounds across both datasets. However, this bound tightening does not correspond to minimal deception unless the strong model is also trained with reinforcement learning. This highlights that minimizing the weak-to-strong misfit bound alone is neither necessary nor sufficient for reducing deceptive behavior.

Examining the decomposition further reveals that bias terms for both weak and strong models are negligible, and variances remain nearly constant within each dataset. As a result, differences in the alignment bound are dominated by covariance terms, particularly the covariance between the weak and strong reward models. Pipelines with higher deception consistently exhibit stronger positive covariance, indicating that deception arises when the strong model amplifies or preserves the weak model's reward misalignment. Reinforcement learning on the strong model substantially reduces deception by disrupting this covariance, even when the overall alignment bound increases.

Finally, while the exact deception minimum varies across datasets, the underlying mechanism is consistent. Supervised fine-tuning of the strong model tends to propagate weak-model inductive biases, whereas reinforcement learning actively reshapes the reward landscape to decorrelate weak and strong predictions, thereby suppressing deception. These results suggest that controlling covariance alignmen rather than merely minimizing bias, variance, or upper bounds is important to prevent deception in weak-to-strong alignment.

## 7 CONCLUSIONS

In this work, we analyzed weak-to-strong (W⇒S) alignment through a bias-variance perspective across multiple training pipelines. We theoretically derived a misfit-based upper-bound on the weak to strong alignment, and related it to the bias-variance and covariance terms. Also by empirically evaluating SFT, RLHF, and RLAIF-based W⇒S variants on PKU-SafeRLHF and HH-RLHF, we connected changes in bias, variance, and covariance to the emergence of deceptive behavior.

Our results show that deception is not primarily governed by bias or variance, which remain small or nearly constant across training pipelines, but instead by covariance alignment between weak and strong reward models. Pipelines that exhibit high deception consistently show strong positive covariance, indicating that the strong model amplifies the weak model's inductive errors rather than correcting them. In contrast, reinforcement learning applied to the strong model substantially reduces deception by disrupting this covariance, even in regimes where the theoretical weak-to-strong misfit bound increases.

Importantly, we find that minimizing the weak-to-strong alignment bound is neither necessary nor sufficient for minimizing deception. While supervised fine-tuning of the weak model yields the tightest theoretical bounds, it does not guarantee low deception unless the strong model is also optimized using reinforcement learning. Conversely, pipelines with larger alignment bounds can achieve minimal deception when covariance alignment is suppressed. This highlights a fundamental limitation of relying solely on misfit-based bounds as safety indicators in weak-to-strong alignment settings.

These findings suggest that safe weak-to-strong alignment requires controlling the interaction between weak and strong models, rather than only improving the fidelity of the weak teacher. From a practical perspective, our analysis supports the use of reinforcement learning objectives at the strong model stage to actively reshape reward landscapes and mitigate the propagation of weak-model biases. More broadly, our framework provides a principled lens for diagnosing and predicting deceptive behavior through bias-variance-covariance decomposition.

Future work may extend this analysis to additional alignment objectives, larger model scales, and interactive or online settings, as well as explore algorithmic interventions that directly regularize covariance alignment. We believe that such directions are essential for developing robust weak-to-strong alignment pipelines that remain reliable even when weak supervision is imperfect.

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

# APPENDIX

## A  DATASET DETAILS

- **HH-RLHF** Bai et al. (2022a) consists of helpful and harmless subsets, along with red-teaming data. We select 15,000 samples from the harmless subset and partition them into three disjoint splits of 5,000 samples each, used for weak teacher training, strong student training, and held-out evaluation, respectively.
- **PKU-SafeRLHF** Dai et al. (2024) contains paired responses annotated for safety and helpfulness. We select 15,000 samples by choosing the safer response for each prompt and divide them into three splits of 5,000 samples for our training and evaluation pipeline.

## B  EVALUATION DETAILS

### B.1  ESTIMATING BIAS, VARIANCE, AND COVARIANCE FROM REWARD ACCURACIES

We estimate the bias, variance, and covariance terms in Eqs. equation 18–equation 22 using *reward accuracies* rather than KL divergences. Concretely, for each datapoint $x$ (a preference prompt) we treat the reward as a binary indicator of whether a model selects the same option as the human-labeled choice.[1] We estimate expectations over $a \sim \pi^w$ by repeatedly sampling completions from the weak policy and re-running the relabeling/evaluation pipeline.

**Step 1: Estimating the mean reward terms.**  Fix a prompt $x$ and run the weak relabeling procedure for N i.i.d. trials. In trial $n$, the weak policy $\pi^w$ produces an output (action) $a^{(n)} \sim \pi^w(\cdot \mid x)$. We then define three empirical correctness indicators:

- **Ground-truth correctness under weak actions.** Let $r^{*(n)} \triangleq r^*(x, a^{(n)}) \in \{0, 1\}$ indicate whether the weak-sampled output agrees with the original human label. Since $r^*$ is one-hot with respect to the dataset label, we have $\bar{r}^* = \mathbb{E}_{a \sim \pi^w}[r^*]$ equal to the weak model's accuracy on the ground-truth labels.
- **Weak self-consistency on weak-labeled data.** Let $\hat{r}^{\pi^w,(n)} \triangleq \hat{r}^{\pi^w}(x, a^{(n)}) \in \{0, 1\}$ be the indicator that a *fresh* weak-model judgment agrees with the weak-generated pseudo-labels (i.e., we relabel the dataset with the weak model, then re-evaluate the weak model against those labels).
- **Strong agreement with weak-labeled data.** Let $\hat{r}^{\pi^s,(n)} \triangleq \hat{r}^{\pi^s}(x, a^{(n)}) \in \{0, 1\}$ be the indicator that the strong model's judgment agrees with the weak pseudo-labels when evaluated on the same weak-labeled set.

---

[1]Equivalently, the ground-truth reward $r^*(x, a) \in \{0, 1\}$ is a one-hot correctness indicator; thus its accuracy is 1 on the human-labeled answer.

Using these indicators, we estimate the mean reward terms by sample averages:

$$\bar{r}^* \approx \frac{1}{N} \sum_{n=1}^{N} r^{*(n)}, \tag{26}$$

$$\bar{r}^{\pi^w} \approx \frac{1}{N} \sum_{n=1}^{N} \hat{r}^{\pi^w,(n)}, \tag{27}$$

$$\bar{r}^{\pi^s} \approx \frac{1}{N} \sum_{n=1}^{N} \hat{r}^{\pi^s,(n)}. \tag{28}$$

All three expectations are taken with respect to $a \sim \pi^w$; empirically, this corresponds to using the weak policy as the sampling distribution and re-running the corresponding evaluation procedure N times.

**Step 2: Estimating bias, variance, and covariance.**  Given the empirical means above, we compute the squared bias terms exactly as in Eqs. equation 18–equation 22:

$$\text{Bias}^2(\hat{r}^{\pi^w}) \triangleq (\bar{r}^{\pi^w} - \bar{r}^*)^2, \tag{29}$$

$$\text{Bias}^2(\hat{r}^{\pi^s} \mid \hat{r}^{\pi^w}) \triangleq (\bar{r}^{\pi^s} - \bar{r}^{\pi^w})^2. \tag{30}$$

For the remaining second-order terms, we estimate variances/covariances by sample moments under the same weak-action distribution:

$$\text{Var}(\hat{r}^{\pi^w}) \approx \frac{1}{N} \sum_{n=1}^{N} (\hat{r}^{\pi^w,(n)} - \bar{r}^{\pi^w})^2, \tag{31}$$

$$\text{Var}(r^*) \approx \frac{1}{N} \sum_{n=1}^{N} (r^{*(n)} - \bar{r}^*)^2, \tag{32}$$

$$\text{Cov}(r^*, \hat{r}^{\pi^w}) \approx \frac{1}{N} \sum_{n=1}^{N} (r^{*(n)} - \bar{r}^*)(\hat{r}^{\pi^w,(n)} - \bar{r}^{\pi^w}), \tag{33}$$

$$\text{Var}(\hat{r}^{\pi^s}) \approx \frac{1}{N} \sum_{n=1}^{N} (\hat{r}^{\pi^s,(n)} - \bar{r}^{\pi^s})^2, \tag{34}$$

$$\text{Cov}(\hat{r}^{\pi^s}, \hat{r}^{\pi^w}) \approx \frac{1}{N} \sum_{n=1}^{N} (\hat{r}^{\pi^s,(n)} - \bar{r}^{\pi^s})(\hat{r}^{\pi^w,(n)} - \bar{r}^{\pi^w}). \tag{35}$$

**Reporting the misfit bound (including covariances).**  Finally, we report all terms in Eqs. equation 18–equation 22 and the corresponding bound on the misfit that depends on the covariance terms. In our setting, the bound is computed by substituting the empirical estimates of $\text{Bias}^2(\hat{r}^{\pi^w})$, $\text{Bias}^2(\hat{r}^{\pi^s} \mid \hat{r}^{\pi^w})$, $\text{Var}(r^*)$, $\text{Var}(\hat{r}^{\pi^w})$, $\text{Var}(\hat{r}^{\pi^s})$, $\text{Cov}(r^*, \hat{r}^{\pi^w})$, and $\text{Cov}(\hat{r}^{\pi^s}, \hat{r}^{\pi^w})$ into the misfit inequality in 25.

### B.2  CONFIDENCE SCORE

For a pair of outputs $a^1 \succ a^2$ associated with prompt $x$, we define the confidence score of the model $\pi$ as:

$$\mathcal{C}^\pi(x) = \sigma(\ell_\pi(a^1|x) - \ell_\pi(a^2|x)), \tag{36}$$

where $\ell^\pi(a|x)$ denotes the average log probability of completion $a$ under model $\pi$:

$$\ell_\pi(a|x) = \frac{1}{|a|} \sum_{i=1}^{|a|} \log \pi(t_i|x). \tag{37}$$

The sigmoid function $\sigma(\cdot)$ maps the log-probability difference into $[0, 1]$, producing a normalized confidence score indicating how strongly $\pi$ prefers the winning completion over the losing one.

### B.3  DECEPTION SCORE

Following Yang et al. (2025) we define the deception score as cases where the strong model appears highly confident ($\mathcal{C}^{\pi^s}(x) > \tau$) yet produces an incorrect answer ($y^{\pi^s} \neq y^{gt}$), while the weak model remains low-confidence ($\mathcal{C}^{\pi^w}(x) < \tau$) on the same prompt. We sweep $\tau$ but due to calibration/confidence differences across PPO/SFT pipelines, deceptive cases vanish above $\tau = 0.49$. We therefore report $\tau = 0.49$ as the highest threshold with non-zero deceptive cases, and include additional results at $\tau = 0.45$ in Appendix **??**.

$$d = \frac{|x : y^{\pi^s} \neq y^{gt}; \mathcal{C}^{\pi^s}(x) > \tau; \mathcal{C}^{\pi^w}(x) < \tau|}{|x : y^{\pi^s} \neq y^{\text{gt}}|}. \tag{38}$$

## C  IMPLEMENTATION DETAILS

Our weak-to-strong (W$\Rightarrow$S) framework consists of a single weak teacher and a single strong student model. We use Llama-3.2-3B-Instruct to train the weak model and Llama-3.1-8B-Instruct to train the strong model.

During both training and inference, we adopt a simple instruction–response format for single-turn conversations, where user inputs are prefixed with `User:` and model outputs with `Assistant:`. No multi-turn context or additional role conditioning beyond these prefixes is used. We use a learning rate of $2 \times 10^{-5}$ for reward modeling, $1 \times 10^{-5}$ for policy optimization, and $1 \times 10^{-5}$ for supervised fine-tuning (SFT). Batch sizes are set to 4 for reward modeling, 8 for policy training, and 2 for SFT. Hyperparameters are selected based on pilot experiments.

We do not employ LoRA or other parameter-efficient fine-tuning techniques. All experiments are conducted on 1 to 4 NVIDIA A100 GPUs with 80 GB of memory.

After training the weak model, we use it to relabel preference data by prompting it with a fixed comparison template together with the original dataset text. For each instance, the model is asked to select the preferred option and reject the alternative, producing a binary preference label. This procedure converts the original samples into weakly supervised preference pairs, which are subsequently used for training downstream components.

## D  REPRODUCIBILITY STATEMENT

All reported results are obtained by averaging over three independent runs with different random seeds. To support reproducibility and facilitate further analysis, we release the full codebase used in our experiments at $--$.

## E  LIMITATIONS

Our study evaluates a limited configuration consisting of two closely related model sizes from the same model family. We do not employ parameter-efficient fine-tuning methods, and analyzing their impact on the proposed pipeline is left for future work. Moreover, our evaluation focuses exclusively on harmlessness as the alignment objective, whereas practical alignment settings often involve multiple objectives and trade-offs.

We do not explicitly assess robustness to reward hacking or other forms of specification issues. Extending the evaluation to more challenging domains, such as mathematical reasoning Luo et al. (2025); Yang et al. (2024) and code generation Luo et al. (2024); Liu et al. (2023), is an important direction for future work. In addition, we plan to investigate the role of uncertainty in both the weak teacher and strong student models, following recent theoretical and empirical analyses Kausik et al. (2025); Lang et al. (2024); Lou et al. (2024). Finally, broader evaluation across additional model families and datasets remains an important avenue for future study.

