# OpenReview forum: "Evaluating Risks in Weak-to-Strong Alignment: A Bias-Variance Perspective"
_ICLR.cc/2026/Workshop/AFAA — Submitted to AFAA 2026_

### Official Review · Reviewer_k5bA · 2026-02-21
**Evaluating risks in weak to strong alignment.**

**Rating:** 2
**Confidence:** 4

**Summary:**

Authors introduce an empirically studied where four different weak-to-strong training pipelines, spanning SFT, RLHF, and RLAIF , where essentially they construct a pipeline where they first train the weak model through RLHF or SFT and then subsequently generate a training data set from the weak model that can be used to train the strong model. They they formulate the definition of off policy population risk in terms of the reward discrepancies between the strong model and the weak model compared to the ground truth, and establish an upper bound on the off policy population risk of weak to strong generalization.

They set up four pipelines.
1. They train the small model with RLHF  and then it generates D2 a dataset of prompt, label labelled by the weak model. This data set is then used to train the strong model with RLAIF.
2. They train the small model with RLHF  and then it generates D2 a dataset of prompt, label labelled by the weak model. This data set is then used to train the strong model with SFT.
3. They train the small model with SFT human preferred labels from particular datasets (HH-RLHF, and SafeRLHF), and then it generates D2 a dataset of prompt, label labelled by the weak model. This data set is then used to train the strong model with RLAIF.
4. They train the small model with SFT human preferred labels from particular datasets (HH-RLHF, and SafeRLHF), and then it generates D2 a dataset of prompt, label labelled by the weak model. This data set is then used to train the strong model also with SFT.

They calculate Deception scores, where deception  effectively the proportion of times, weak model being not confident in its response, but the strong model being highly confident while being incorrect. They provide observations of deception scores for each of the pipelines, as well as the threshold of empirical risk, which they calculate. They show that when the second set up (RLHF-SFT), results in the highest deception score across data sets and thresholds. They also find that training the weak model with RL consistently increases the upper bound for offpolicy population risk.

**Strengths:**

- They derive and formulate off policy population risk, as well as all the remaining theory behind how they measure bias and variance in the reward scoring mechanisms.
- They conduct experiments across different pipelines, which provides us a comparison of how reinforcement learning and supervised learning might have an effect on weak to strong alignment.

**Weaknesses:**

- The authors do not provide much information on what kinds of models (LLMs) they are using for SFT and RLHF, and how their observations might change across models

- The authors only tested the hypothesis on limited benchmarks, various benchmarks which measure alignment, and it would be interesting to see how the empirical risk compares across different data sets. Further, we see that doing RLHF on the weak model increases the threshold of the policy population. In this, it would be interesting to explore what exactly causes this drift.

- Although the paper covers these different pipelines, it doesn't go into detail about the formulation or use of different RL algorithms and the influence of distribution shift (under diff loss functions) when doing weak-to-strong alignment.

---

### Official Review · Reviewer_cu4W · 2026-02-21
**A well structured theoretical analysis of weak-to-strong alignment that highlights covariance as a driver of deception, but would benefit from clarifications and stronger grounding.**

**Rating:** 3
**Confidence:** 4

**Summary:**

This paper studies weak-to-strong alignment through a bias/variance/covariance decomposition of misfit risk bounds. The authors derive a misfit upper bound on weak-to-strong population risk and decompose it into bias, variance, and covariance terms. They empirically evaluate four alignment pipelines (SFT to SFT, SFT to RLAIF, RLHF to SFT, RLHF to RLAIF) on two datasets with two variants of the Llama model using deception scores as the main metric. Their central claim is that covariance alignment between weak and strong reward models, rather than bias or variance alone, explains model misalignment behaviors.

**Strengths:**

- The paper presents a very interesting and novel work that builds directly on recent weak-to-strong PAPERs STUDYING THIS THEORY. The paper provides a strong theoretical lens and developments rather than purely empirical comparisons. For a workshop paper, the theoretical development is substantial and well-integrated with related work.
- The empirical results show that bias is negligible and variance is nearly constant across the various training settings, datasets and the llama family, while covariance influences deception meaningfully. This is a non-trivial empirical insight and a strong interpretive contribution that explains why stronger models might misalign with their teacher training models.
- The paper presents an explicit framework for linking theoretical quantities, such as variance and bias, to explicit behavioral metrics such as deception.

**Weaknesses:**

- The structure of the article is unclear. The introduction does not clearly explain the principles of weak-to-strong alignment or situate the work within prior literature (no references in the introduction), and it lacks foundational references, making the overall contribution difficult to contextualize. Moreover, the related work section is loosely structured, and while Xu et al. (2025) is cited, the paper does not articulate how its bias/variance/covariance perspective differs from or extends that prior analysis.
- Article accessibility. Key concepts such as “misfit,” “weak-to-strong population risk,” and the deception metric are introduced formally but not explained, making the theoretical framework difficult to follow and limiting accessibility for readers outside the immediate subfield.
- Results Section Clarity. The results section presents detailed numerical decompositions in Table 1, but the connection between specific table values (e.g., covariance terms) and the various results section claims is not explicit, making it difficult to understand which precise numbers support these claims. Moreover, some claims are overstated, in my opinion, especially the covariance effect, as the authors test only Llama-3.2-3B and Llama-3.1-8B, making it unclear whether this behavior would generalize to other model families.
- Relation to fairness & Safety. Although the paper addresses alignment and safety, it does not discuss implications for algorithmic fairness or bias amplification, making the fit with a fairness-focused workshop unclear. I would recommend the author to reduce the theoretical section pushing some demonstration to the appendices and favoring meaningful discussions on the implications of your results in term of safety and fairness to address the community.


Additional Question:  The theoretical derivations rely on expectations over policy distributions and reward-based risk definitions. However, to my understanding, supervised fine-tuning (SFT) does not explicitly optimize a reward objective. Therefore, could the author clarify how the misfit framework and associated assumptions extend to their SFT pipelines?

---

### Official Review · Reviewer_t97W · 2026-02-24
**EVALUATING RISKS IN WEAK-TO-STRONG ALIGN- MENT: A BIAS-VARIANCE PERSPECTIVE**

**Rating:** 2
**Confidence:** 4

**Summary:**

The paper examines risks in weak-to-strong alignment pipelines using a bias, variance, and covariance decomposition. It bounds the strong model off-policy risk using the weak model population risk and teacher-student misfit. The authors evaluate four pipelines on the PKU-SafeRLHF and HH-RLHF datasets. The empirical results state that deception correlates with covariance alignment between weak and strong reward models, and that training the strong model with reinforcement learning reduces deception by altering this covariance.

**Strengths:**

1. Applying a bias, variance, and covariance decomposition to weak-to-strong alignment provides a specific, measurable framework for evaluating deception risk.
2. The experimental setup tests four standard alignment pipelines (RLHF to SFT, RLHF to RLAIF, SFT to SFT, SFT to RLAIF) across two established safety datasets.
3. The hypothesis links covariance structure to deceptive behavior, isolating a testable mechanism for alignment failure.

**Weaknesses:**

1. The empirical estimation of bias, variance, and covariance relies on binary accuracy indicators rather than continuous reward predictions. This substitution distorts second-order statistics and breaks the mathematical link to the squared-loss derivations.
2. The experimental design lacks required ablations. There is no sensitivity analysis for the number of weak-policy samples, reinforcement learning parameters.
3. The evaluation omits empirical comparisons to established baselines that mitigate weak-label noise, such as auxiliary confidence losses, reliability-aware filtering, or contrastive weak-to-strong methods.

---

### Meta-Review · Area_Chair_nfnD · 2026-02-26

**Recommendation:** Reject
**Confidence:** 3

**Metareview:**

The paper investigates weak-to-strong alignment by decomposing bias-variance-covariance to analyze the risks of training strong models with imperfect supervision. It derives a misfit-based upper bound on population risk and argues that deception is primarily driven by the covariance between the weak teacher and strong student. While the theoretical framework attempts to bridge the gap between abstract risk theory and practical post-training pipelines, reviewers found the work underdeveloped and lacking the necessary depth for a workshop or conference track.

The decision for rejection is based on the following key points:

* **Insufficient Novelty and Impact:** Reviewers noted that the application of bias-variance decomposition to this setting felt incremental and did not provide enough new insight beyond existing literature on weak-to-strong generalization.
* **Technical and Empirical Gaps:** The submission lacks a rigorous exploration of different RL algorithms and fails to sufficiently analyze how distribution shifts under various loss functions influence the alignment outcome.
* **Limited Scope of Evaluation:** The empirical results are restricted to a single model family and a narrow alignment objective (harmlessness), leaving questions about the framework's generalizability to more complex reasoning tasks or diverse architectures.
* **Lack of Clarity:** Critical sections regarding the formulation of the risk bounds and their practical implications were found to be unclear, hindering the overall quality and soundness of the manuscript.

---

### Decision · Program_Chairs · 2026-03-02

Reject